# Ten-Eleven Translocation 1 and 2 Enzymes Affect Human Skin Fibroblasts in an Age-Related Manner

**DOI:** 10.3390/biomedicines11061659

**Published:** 2023-06-07

**Authors:** Paulina Kołodziej-Wojnar, Joanna Borkowska, Anna Domaszewska-Szostek, Olga Bujanowska, Bartłomiej Noszczyk, Natalia Krześniak, Marek Stańczyk, Monika Puzianowska-Kuznicka

**Affiliations:** 1Department of Geriatrics and Gerontology, Medical Centre of Postgraduate Education, 01-813 Warsaw, Poland; 2Department of Human Epigenetics, Mossakowski Medical Research Institute, PAS, 02-106 Warsaw, Poland; 3Department of Plastic Surgery, Medical Centre of Postgraduate Education, 01-813 Warsaw, Poland; noszczyk@melilot.pl (B.N.);; 4Department of General and Oncological Surgery with Traumatic Unit, Wolski Hospital, 01-211 Warsaw, Poland; 5Faculty of Medicine, Lazarski University, 02-662 Warsaw, Poland

**Keywords:** human primary fibroblasts, ten-eleven translocation methylcytosine dioxygenase 1 (TET1), ten-eleven translocation methylcytosine dioxygenase 2 (TET2), intrinsic aging, proliferation, apoptosis, autophagy, DNA damage and repair

## Abstract

Ten-eleven translocation (TET) enzymes catalyze the oxidation of 5-methylcytosine (5mC), first to 5-hydroxymethylcytosine (5hmC), then to 5-formylcytosine (5fC), and finally to 5-carboxycytosine (5caC). Evidence suggests that changes in TET expression may impact cell function and the phenotype of aging. Proliferation, apoptosis, markers of autophagy and double-strand DNA break repair, and the expression of Fibulin 5 were assessed by flow cytometry in TET1 and TET2-overexpressing fibroblasts isolated from sun-unexposed skin of young (23–35 years) and age-advanced (75–94 years) individuals. In cells derived from young individuals, TET1 overexpression resulted in the inhibition of proliferation and apoptosis by 37% (*p* = 0.03) and 24% (*p* = 0.05), respectively, while the overexpression of TET2 caused a decrease in proliferation by 46% (*p* = 0.01). Notably, in cells obtained from age-advanced individuals, TETs exhibited different effects. Specifically, TET1 inhibited proliferation and expression of autophagy marker Beclin 1 by 45% (*p* = 0.05) and 28% (*p* = 0.048), respectively, while increasing the level of γH2AX, a marker of double-strand DNA breaks necessary for initiating the repair process, by 19% (*p* = 0.04). TET2 inhibited proliferation by 64% (*p* = 0.053) and increased the level of γH2AX and Fibulin 5 by 46% (*p* = 0.007) and 29% (*p* = 0.04), respectively. These patterns of TET1 and TET2 effects suggest their involvement in regulating various fibroblast functions and that some of their biological actions depend on the donor’s age.

## 1. Introduction

An inherent feature of aging is epigenetic drift [1,2], which involves the progressive accumulation of changes in the epigenome resulting in unwanted changes in gene expression patterns [3]. Consequently, epigenetic drift contributes to impaired functions of aging cells [2,4,5]. 

Regulation of the epigenetic code involves ten-eleven translocation methylcytosine dioxygenase (TET) enzymes. Three TET enzymes (TET1-3) have been identified [6]. Although all are expressed in various human tissues, their expression levels differ according to tissue type ([7], http://www.proteinatlas.com, accessed on 10 November 2022). TET enzymes catalyze the oxidation of 5-methylcytosine (5mC), first to 5-hydroxymethylcytosine (5hmC), then to 5-formylcytosine (5fC), and finally to 5-carboxycytosine (5caC) [8,9]. 5hmC enrichment is detected within gene bodies, promoters, and transcription factor binding sites [10,11]. It can support gene activation by engaging specific proteins that modify chromatin structure [12,13]. 

Little is known about the involvement of TETs in aging in various organs. An age-related decline in TET1 expression and a decrease in the global level of 5hmC have been observed in human peripheral blood mononuclear cells [14]. In contrast, in the mouse hippocampus, 5hmC content was shown to increase during aging without changes in TETs expression [15]. However, it is unknown whether the action of TET1 on cell function remains the same regardless of age. In turn, aging is known to be associated with the increased frequency of somatic mutations of the TET2 gene and an increased risk of hematological pathologies [16,17]. In addition, it has been shown that, in human cells and tissues, as well as in mouse models, loss of TET2 is accompanied by the increased severity of vascular injury and atherosclerosis [18,19]. In the hippocampal neurogenic niche of adult mice, overexpression of Tet2 prevented an age-related decline in neurogenesis and enhanced learning and memory capabilities [20]. However, little information has been obtained regarding the involvement of TETs in the aging of other tissues and organs, including the skin.

We previously showed that the expression of TETs at either the mRNA or the protein levels in dermal fibroblasts and adipose stem cells (ASCs) is similar between young and age-advanced individuals [21,22]. However, the specific activities of TETs may change in response to aging-related changes in the cellular environment, such as the accessibility of various cofactors. Indeed, we have identified a number of age-associated differentially hydroxymethylated regions in fibroblasts and ASCs [21,22]. Moreover, as indicated by other authors, TET enzymes affect gene transcription not only by demethylating DNA but also by mechanisms that rely on their interaction with other proteins [10,23,24]. Therefore, despite the lack of age-associated differences in expression, TET1 and TET2 may affect cell function in an age-related manner. To test this hypothesis, we overexpressed these enzymes in human dermal fibroblasts obtained from young and age-advanced donors since such cells more closely resemble natural aging than cells aged in vitro [25]. Thus, such studies provide a better understanding of epigenetic drift and the function of TETs in aging human skin and might contribute to the development of strategies to delay the aging process.

## 2. Materials and Methods

### 2.1. Fibroblast Isolation

Healthy skin samples (2 cm^2^) from a sun-unexposed area of the suprapubic region were collected during elective surgery performed on 5 young (23–35 years) and 5 age-advanced (75–94 years) women without acute inflammation, free of local and systemic diseases possibly affecting the skin’s condition such as diabetes, cancer, or autoimmunity. Upon collection, samples were placed in 40 mL of ice-cold phosphate-buffered saline and immediately transported to the laboratory. Primary fibroblast cultures were established by enzymatic digestion in accordance with the protocol described by Rittié and Fisher [26] with the following modifications: to separate epidermis from dermis, skin samples were digested overnight at 4 °C with dispase II (Sigma-Aldrich, St. Louis, MO, USA). Next, the dermis was digested overnight at 37 °C with collagenase IV (Sigma-Aldrich, St. Louis, MO, USA). For functional experiments, cells from passages 4 or 5 were used.

### 2.2. Preparation of TET-Expressing Vectors

A DNA fragment encoding TET1 was amplified from the FH_TET1-pEF vector (a kind gift from Dr. Anjana Rao [27]) with forward primer 5′-**CACCGGCG**ATGTCTCGATCCCGCCATG-3′ (the restriction site for the SgrAI enzyme is shown in bold, the ATG codon is underlined) and reverse primer 5′-**GCTAGC**TCAGACCCAATGGTTATAGGG-3’ (the restriction site for the NheI enzyme is shown in bold, the stop codon is underlined). The TET2-encoding sequence was amplified from 25 ng of cDNA prepared on the template of mRNA isolated from the human dermis, with forward primer 5′-**GGCGCC**ATGGAACAGGATAGAACCAAC-3′ (the restriction site for the KasI enzyme is shown in bold, the ATG codon is underlined) and reverse primer 5′-**AGCGCT**TCATATATATCTGTTGTAAGGC-3′ (the restriction site for the Eco47III enzyme is shown in bold, the stop codon is underlined). Amplification was performed using the Phusion High-Fidelity PCR Kit (Thermo Fisher Scientific, Waltham, MA, USA). The PCR reaction was as follows: 30 s at 98 °C; 10 cycles of 10 s at 98 °C, 30 s at 62 °C, and 4 min 30 s at 72 °C; 25 cycles of 10 s at 98 °C and 4 min 50 s at 72 °C and final extension for 5 min at 72 °C. Next, the TET-encoding DNA fragments were cloned into the pJET1.2/blunt vector from the CloneJET PCR Cloning Kit (Thermo Fisher Scientific, Waltham, MA, USA), cut with the respective enzymes, and re-cloned into the pSELECT-GFPzeo plasmid (InvivoGen, San Diego, CA, USA). Cloning fidelity was verified by restriction analysis.

### 2.3. Cell Transfection

Fibroblasts or HEK293 cells were seeded onto six-well plates at 0.9 × 10^5^ cells/well and cultured for 24 h in Dulbecco’s Modified Eagle’s Medium, with high glucose, GlutaMAX™ Supplement, HEPES (Gibco, Waltham, MA, USA), and 10% heat-inactivated fetal bovine serum. Twenty-four hours later, transfection was performed using the K2 Transfection System (Biontex, Munich, Germany), in accordance with the manufacturer’s protocol, with 3.8 µg per well of pSELECT-GFPzeo-TET1 plasmid encoding TET1 or pSELECT-GFPzeo-TET2 encoding TET2. The transfection medium was replaced with fresh medium 6 h later. Cells were collected and analyzed 48 h after transfection. Cells transfected with 3.8 µg of “empty” pSELECT-GFPzeo vector served as a control.

For stable transfection, fibroblasts were seeded onto a six-well plates (0.5 × 10^5^ cells/well), cultured and transfected as above. Twenty-four hours after transfection, the medium was replaced with a fresh medium supplemented with 400 µg/mL Zeocin (InvivoGen, San Diego, CA, USA). The selection medium was changed every 72 h. Stable TET1 transfectant lines were obtained after two weeks. No TET2 stable transfectants were produced.

### 2.4. Immunofluorescence

Primary skin fibroblasts stably expressing TET1 were seeded on coverslips placed in a 24-well dish (0.1 × 10^5^ cells/well). Cells attached to the coverslip were fixed in 4% paraformaldehyde, washed, permeabilized with 0.5% Triton X-100, incubated in 4 N HCl, neutralized, blocked with 10% goat serum, 3% BSA, and 0.1% Triton X-100, then incubated overnight at 4 °C with anti-TET1 (1:100, Thermo Fisher Scientific, Waltham, MA, USA) or anti-5hmC (1:300, Active Motif, La Hulpe, Belgium) antibodies, washed and then incubated with the appropriate secondary antibodies. Fluorescence was analyzed using the LSM 780/ELYRA PS.1 confocal microscope (Zeiss, Jena, Germany). Laser power and acquisition parameters were similar for each image. The “Range Indicator” option was used, which helps to set the image lighting objectively. There was no post-capture image manipulation. For each sample, 50 nuclei were quantified.

### 2.5. Isolation of Proteins and Immunoblotting

To verify whether transfection with pSELECT-GFPzeo-TET1 or pSELECT-GFPzeo-TET2 vectors results in the production of the respective TET proteins, HEK293 cells were suspended in 500 µL of RIPA buffer containing protease inhibitor mix, incubated on ice for 10 min, and then centrifuged at 400× *g* for 20 min at 4 °C. The protein concentration was determined using the BCA Protein Assay kit (Thermo Fisher Scientific, Waltham, MA, USA). Twenty micrograms of the extract were used for electrophoresis and transferred to the membrane following standard procedures. The membrane was blocked and incubated with anti-TET1 (1:1000; Abcam, Cambridge, UK), anti-TET2 (1:500, Active-Motif, Carlsbad, CA, USA), or anti-GFP (1:1000, Thermo Fisher Scientific, Waltham, MA, USA) primary antibodies, washed in TBST and incubated with the appropriate secondary antibodies. Signals were detected using the Clarity ECL Western Blotting Substrate (Bio-Rad, Hercules, CA, USA) and analyzed using a GeneGnome chemiluminescence imaging system (Syngene, Cambridge, UK) and Image Studio Lite 4.0 software (LI-COR, Lincoln, NE, USA) (Appendix A).

### 2.6. Proliferation, DNA Damage, and Apoptosis Assays

Proliferation was assayed in fixed cells using the APC BrdU Flow Kit (BD Biosciences, San Jose, CA, USA) and DNA damage was assessed by measuring γH2AX using the Apoptosis, DNA Damage, and Cell Proliferation Kit (BD Biosciences, San Jose, CA, USA). Forty-eight hours after transfection, cells were fixed and stained according to the manufacturer’s protocols. Apoptosis was evaluated in unfixed cells using the PE Annexin V assay (BD Biosciences, San Jose, CA, USA). Forty-eight hours after transfection, cells were stained following the manufacturer’s protocol. Next, the transfected cells expressing GFP protein encoded by the pSELECT-GFPzeo backbone and marker of interest were analyzed with a BD FACS Canto II Cytometer and BD FACS Diva Software v. 6.1.3 (BD Biosciences, San Jose, CA, USA). This software allows for the analysis of both expression levels and the percentage of cells expressing a given marker.

### 2.7. Beclin 1 and Fibulin 5 Expression

Forty-eight hours after transfection, cells were fixed with Cytofix/Cytoperm™ (BD Biosciences) and incubated with anti-Beclin 1 (1:200; Abcam, Cambridge, UK) or anti-Fibulin 5 (1:200; Biorbyt, Cambridge, UK) antibodies for 40 min. Next, the cells were incubated with secondary antibodies, as appropriate, for 30 min. The transfected cells expressing GFP protein encoded by the pSELECT-GFPzeo backbone and protein of interest were analyzed with a BD FACS Canto II Cytometer and BD FACS Diva Software v. 6.1.3 (BD Biosciences, San Jose, CA, USA). 

### 2.8. Statistical Analysis

Statistical calculations were performed using the GraphPad Prism 7 software (GraphPad Software, San Diego, CA, USA). To assess the normality of the distribution, the Shapiro-Wilk test was used. The effects of TET overexpression were analyzed by a Mann-Whitney *U* test (young pSELECT-GFPzeo vs. age-advanced pSELECT-GFPzeo) and Wilcoxon matched-pairs signed rank test (pSELECT-GFPzeo vs. pSELECT-GFPzeo-TET1 or pSELECT-GFPzeo-TET2). For mean fluorescence intensity (MFI) quantified with the ZEN blue edition software v. 2.6 (Zeiss, Jena, Germany), statistical analysis was performed with Student’s *t*-test. For all tests, the level of significance was set at 0.05.

## 3. Results

### 3.1. TET Overexpression in Primary Fibroblasts

The efficiency of primary fibroblasts transfection with TET-overexpressing vectors was low, 5–15% (Appendix A). Therefore, to verify whether the transfection of human cells with these vectors results in the production of the encoded proteins, we used HEK 293 cells and found that both TET1 and TET2 were efficiently overexpressed (Appendix A). Next, to verify if increased amounts of TETs in primary fibroblasts result in molecular changes, we checked whether stable overexpression of TET1 in these cells affects DNA 5-hydroxymethylation. We observed a significant increase in the quantity of 5hmC compared to cells expressing only endogenous TET1 (Figure 1), which, supposedly, based on other authors’ data, can affect the regulation of transcription [12,13]. 

### 3.2. Effects of TET1 and TET2 on Proliferation

As the transfection procedure itself and successful absorption of exogenous genetic material can affect cell function [28], to establish the best control for the transfection experiments, we first compared the proliferation of cells that had undergone a transfection procedure but not been transfected with a vector versus cells successfully transfected with the “empty” pSELECT-GFPzeo vector. We found that proliferation was significantly affected by DNA absorption. Therefore, to limit the number of variables that could affect cell functioning, in all subsequent experiments cells successfully transfected with an “empty” pSELECT-GFPzeo vector were used as controls (control fibroblasts).

The proliferation of control fibroblasts obtained from age-advanced individuals was 80% lower (*p* = 0.01) than that of control fibroblasts obtained from younger individuals. After transfection with pSELECT-GFPzeo-TET1, leading to the overexpression of TET1, the proliferation of fibroblasts obtained from young and age-advanced individuals decreased by 37% (*p* = 0.03) and 45% (*p* = 0.05), respectively, compared with that of age-matched control fibroblasts (Figure 2A). Similarly, the overexpression of TET2 resulted in a 46% decrease in the proliferation of fibroblasts originating from young individuals (*p* = 0.01) and a 64% decrease in fibroblasts originating from age-advanced individuals, although the level of significance has not been reached (*p* = 0.053) (Figure 2A).

### 3.3. Effects of TET1 and TET2 on Apoptosis

The apoptosis of control fibroblasts originating from age-advanced individuals was 34% higher than that of cells from young individuals, although this difference only showed a tendency towards significance (*p* = 0.055). After transfection with pSELECT-GFPzeo-TET1, the apoptotic potential of fibroblasts obtained from young individuals significantly decreased by 24% (*p* = 0.05) compared with that of age-matched controls not overexpressing TET1. In cells obtained from age-advanced individuals, there was no change in apoptosis upon TET1 overexpression (Figure 2B). In addition, TET2 overexpression did not affect the level of apoptosis in either age group (Figure 2B).

### 3.4. Effects of TET1 and TET2 on Autophagy Marker Beclin 1

Beclin 1 is a key regulator of autophagy that forms a complex initiating autophagosome formation and maturation. There was no significant age-associated difference in Beclin 1 expression and the percentage of Beclin 1-expressing cells in control fibroblasts.

However, upon TET1 overexpression, in fibroblasts obtained from age-advanced donors, the level of Beclin 1 significantly decreased by 28% (*p* = 0.048). Such an effect was not observed in fibroblasts originating from young donors. In addition, the percentage of cells expressing Beclin 1 did not change significantly in either age group (Figure 3A). Transfection with pSELECT-GFPzeo-TET2 did not affect this protein’s expression or the percentage of cells expressing it (Figure 3B). 

### 3.5. Effects of TET1 and TET2 on Double-Strand DNA Break Repair Marker γH2AX

The level of phosphorylated histone H2AX (γH2AX), a marker of the repair of double-strand DNA breaks, was higher by 27% in control cells originating from age-advanced donors than in control cells from young donors (*p* = 0.02). In contrast, the percentage of control cells from age-advanced study participants showing the presence of γH2AX was lower by 54% (*p* = 0.02) than that of cells from young donors. 

Upon the overexpression of TET1, the level of γH2AX increased by 19% (*p* = 0.04) in cells originating from age-advanced donors, while it remained unaffected in cells from young donors. The percentage of γH2AX-expressing cells did not change in either age group (Figure 4A). The overexpression of TET2 increased the γH2AX expression by 46% (*p* = 0.007) in cells of age-advanced donors, while it did not affect γH2AX levels in cells originating from younger study participants. The percentage of γH2AX-positive cells did not change in either age group (Figure 4B).

### 3.6. The Effect of TET1 and TET2 on Fibulin 5 Expression

Analysis of the expression of Fibulin 5, a protein synthesized and secreted by fibroblasts, revealed no age-associated differences in control cells. The overexpression of TET1 did not influence the level of Fibulin 5 expression or the percentage of cells expressing this protein in both age groups (Figure 5A). In turn, the overexpression of TET2 increased Fibulin 5 expression by 29% (*p* = 0.04) in cells originating from age-advanced donors, without increasing the number of cells expressing this protein. Meanwhile, fibroblasts obtained from young donors remained unaffected by TET2 overexpression (Figure 5B).

## 4. Discussion

Data indicate that TET enzymes regulate the function of embryonic stem cells by maintaining pluripotency and cell fate during development by regulating differentiation [29]. Under physiological conditions, deregulation of the TET enzymes mostly affects stem cells, blood cells, and cells associated with reproduction [30,31]. Notably, epigenetic drift involving aging-associated changes in 5-hydroxymethylation is an inseparable element of growing old. However, even though there are data regarding the aging-related changes in 5hmC [21,32], there is much less data regarding the role of TET enzymes in human aging. For example, the expression of TETs in postmortem brains of Alzheimer’s disease sufferers was significantly decreased compared to age-matched, disease-free control brains [33]. TET3 but not TET1 or TET2 protein expression increased with age in human subcutaneous adipose stem cells [21]. A comprehensive study in short-lived killifish *Nothobranchius furzeri* aging model revealed a substantial reduction in *tet3* transcriptional activity in the brain and liver, coexisting with a significant age-associated decrease in 5hmC, while the expression of TET1 and TET2 remained unchanged [34].

In this work, we analyzed the roles of TET1 and TET2 in dermal fibroblasts originating from young and age-advanced individuals in order to identify functions regulated by these proteins, to determine whether such regulation is age-related, and to clarify whether the increased expression of TETs in cells of age-advanced individuals can improve their function. We showed that the overexpression of TET1 and TET2 exerted a variety of biological effects, some of which seemed to be age-dependent (Table 1).

The inhibition of the proliferation of healthy primary fibroblasts induced by the overexpression of TET1 and TET2 is in agreement with the results of studies on various cancer cell lines [35,36]. However, the opposite findings were obtained in some cases; for example, in hepatocellular carcinoma cells, this was a knockdown of TET2 that inhibited proliferation [37]. Nonetheless, these findings show that TETs are among the key players controlling the proliferation of healthy and cancer cells. Notably, we discovered that the age of cell donors did not affect this anti-proliferative effect in non-cancerous cells. 

The role of TET1 in apoptosis has been the subject of multiple studies involving tumor cell lines. For example, in lines originating from solid tumors such as gastric, colorectal, hepatocellular, and breast tumors, apoptosis increased upon TET1 overexpression [38]. On the other hand, a study of malignant hematopoietic cell lines revealed that apoptosis was induced by TET1 knockout [39]. Although the effect of TET1 on apoptosis in non-tumorigenic cells has not been thoroughly explored, it has been recently shown that the knockdown of this protein in human hepatocyte-like cells promoted this process [40]. In agreement with that previous study, we showed that TET1 overexpression decreased the apoptosis of fibroblasts obtained from young individuals. However, it did not affect fibroblasts obtained from age-advanced study participants. Similarly, data regarding the effects of TET2 on apoptosis are mostly limited to cancer cells. TET2-dependent protection from DNA damage-induced apoptosis has been described in the U-2 OS human osteosarcoma cell line [41]. In contrast, in acute myeloid leukemia, apoptosis was inhibited by TET2 downregulation [42]. In our experiments utilizing normal fibroblasts, we did not observe any effect of TET2 on this process. Therefore, we conclude that the effects of TET1 and TET2 on apoptosis might depend on the age of the cell donor and the cell’s pathophysiological state. 

Autophagy, a process leading to the degradation of nonfunctional organelles and proteins, supports cellular homeostasis. Autophagic activity decreases with age, contributing to cellular damage and senescence. Nonetheless, previous data showed similar expression of the key autophagy protein Beclin 1 in young and aged fibroblasts [43], and our data on control cells obtained from young and age-advanced individuals confirmed this. We also found that, upon TET1 overexpression, the level of Beclin 1 did not change in cells from young donors, but significantly decreased in cells originating from older study participants. In contrast, TET2 did not affect this protein level in both age groups. These findings contrast with previous data showing that the overexpression of TET2 increased Beclin 1 expression in endothelial cells and atherosclerotic plaques [44]. Therefore, it remains to be elucidated whether TET1 and TET2 affect autophagy in a pathophysiological state-dependent and age-related manner.

The accumulation of double-strand DNA breaks plays a significant role in cellular senescence. Such damage results in the rapid phosphorylation of the histone H2AX, which plays a crucial role in signaling, initiating, and facilitating the repair process [45,46]. We showed that the amount of γH2AX was higher in cells from older donors, suggesting more DNA damage and intensive repair. On the other hand, the percentage of cells showing the presence of γH2AX was lower in cultures from age-advanced donors than those from young donors. This in turn suggests that cells originating from age-advanced donors with irreparable DNA damage could have been eliminated. The overexpression of TET1 and TET2 increased the γH2AX level in cells originating from age-advanced study participants, but did not lead to a change in cultured cells originating from young donors. It was also previously shown in a study on the involvement of Tet2 and Tet3 in the pathogenesis of myeloid cancer in mice that Tet2 knockout did not affect the level of γH2AX [47]. However, in the report on that study, the animals were described only as “adults”. We assume that they were not old because old age is usually explicitly stated. If this assumption is correct, our results showing the lack of an effect of TET2 on γH2AX in cells from young humans could be consistent with this finding. On the other hand, TET1-deficient human glial cells displayed lower levels of γH2AX following irradiation [48]. Here, we hypothesize that TET1 and TET2 promote DNA damage recognition and repair in cells of age-advanced individuals, preventing them from entering a senescent state or facilitating the removal of such cells. 

Changes in fibroblasts’ functional status contribute to impaired wound healing and visible signs of aging. Fibroblasts produce extracellular matrix components, including fibronectin, collagen, and elastin, which are secreted from the cell [49]. However, since we could reach only 5–15% TET transfection efficiency in our primary cultures, we would most likely not have detected a significant difference in their levels in cell culture supernatants. Data from other researchers have shown that Fibulin 5, a glycoprotein involved in the induction of elastic fiber assembly and maturation, can be a useful marker of skin aging. Its levels were shown to decrease with age in fibroblasts, and its loss was associated with the loss of elastic fibers in the skin and carotid arteries [25,50,51]. Moreover, it can be easily detected intracellularly and, therefore, is suitable for flow cytometry evaluation. We showed that TET1 but not TET2 increased the expression of Fibulin 5 in cells from individuals of advanced age. None of the TETs altered the percentage of Fibulin 5-expressing cells in either age group. As there are no other data regarding the relationship between TETs and Fibulin 5, it remains to be confirmed in other experimental settings whether and how TETs affect Fibulin 5 expression in an age-related manner.

Our work has some limitations. First, to obtain cell numbers sufficient for all experiments, we had to perform short-term cell cultures, which could have resulted in culturing-related functional changes or replicative senescence. Second, we analyzed a small number of cell lines. However, they were very homogeneous, as we used very restrictive inclusion criteria: donors were free of local and generalized diseases that could have affected skin condition, the skin fragments originated from the same region of sun-unexposed parts of the body, and the age difference between young and age-advanced donors was large ranging from 40 to 61 years. Third, transfection efficiency was relatively low. However, we found no consistent differences between transfection efficiency with TET1- and TET2-encoding vectors or age-related differences; the differences were random and varied experiment-to-experiment (Appendix A). Fourth, we studied only cells originating from females. Nonetheless, we speculate that the effects of TET1 and TET2 in male cells could be similar, as we found no age-related differences in their mRNA and protein expression levels in fibroblasts from males and females [22]. Nevertheless, the potential gender-specific differences in TETs functions in aging need further investigation. Finally, this study merely describes the effects of increased expression of TETs without analyzing the molecular mechanisms behind them. 

## 5. Conclusions

To our knowledge, this is the first comprehensive study assessing the effects of TET1 and TET2 on a set of cellular functions in human fibroblasts originating from young and age-advanced individuals (Table 1). These two proteins have some similar and distinct biological effects depending on the developmental stage, tissue type, and pathophysiological state. We show here that some of TETs’ functions might also depend on age which, most likely, does not result from changes in TETs expression or catalytic activity, as we did not detect them [22]. However, the catalytic activity was evaluated for all TET enzymes together [22], and it cannot be ruled out that the activity of a single TET indeed changes with age. Based on the available data by other authors, we hypothesize that age-related changes in the biological function of TETs are due to changes in the cell’s redox state, availability of cofactors, or a shift in the expression or activities of TET-interacting proteins [52,53]. We also hypothesize that the increased expression or function of TET1 and TET2 in cells of age-advanced individuals leads to the inhibition of proliferation, improved capacity to correct double-strand breaks, and deceleration of senescence.

## Figures and Tables

**Figure 1 biomedicines-11-01659-f001:**
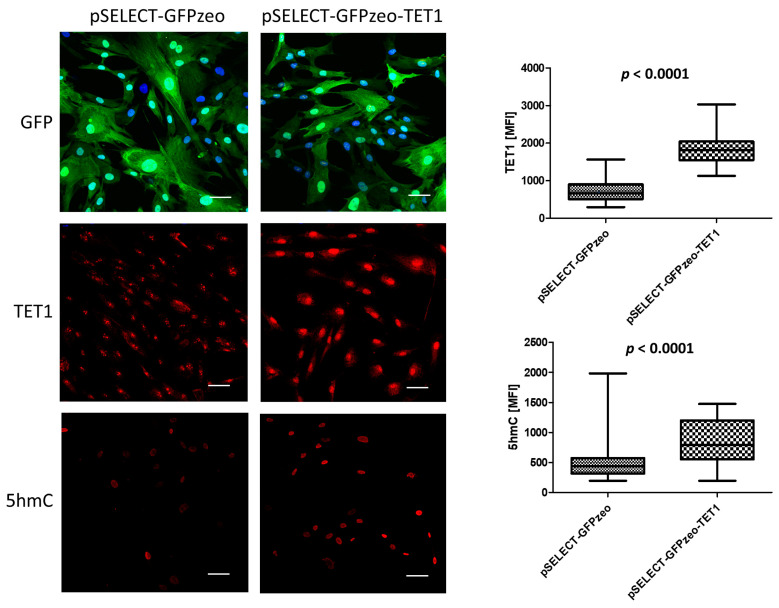
Increased expression of TET1 and amount of 5hmC in primary fibroblasts stably transfected with the pSELECT-GFPzeo-TET1 vector as compared to primary fibroblasts stably transfected with the pSELECT-GFPzeo control vector. Immunofluorescence: GFP—green fluorescent protein, green. TET1—predominantly nuclear localization of TET1, red. 5hmC—nuclear localization of 5-hydroxymethylcytosine, red. Nuclei stained with Hoechst 33342, blue. MFI—mean fluorescence intensity. Scale bar—50 µm.

**Figure 2 biomedicines-11-01659-f002:**
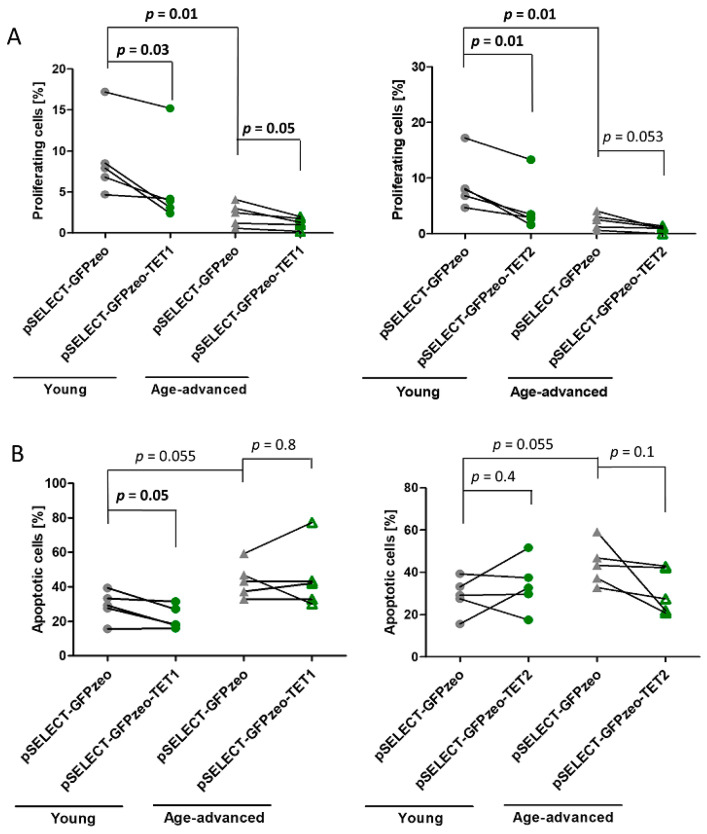
Effects of TET1 and TET2 overexpression on the proliferation and apoptosis of fibroblasts obtained from young (*n* = 5) and age-advanced (*n* = 5) study participants. (**A**). Changes in the percentage of proliferating fibroblasts. (**B**). Changes in the percentage of apoptotic fibroblasts. Statistical analysis was performed using the Mann-Whitney *U* test (young pSELECT-GFPzeo vs. age-advanced pSELECT-GFPzeo) and Wilcoxon matched-pairs signed rank test (pSELECT-GFPzeo vs. pSELECT-GFPzeo-TET1 or pSELECT-GFPzeo-TET2).

**Figure 3 biomedicines-11-01659-f003:**
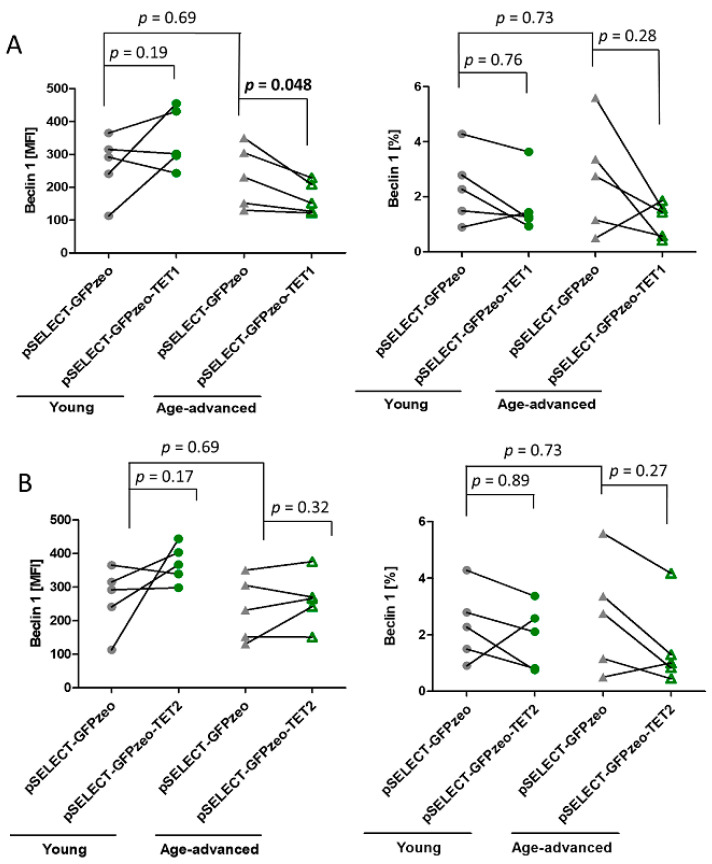
Effects of TET1 and TET2 overexpression on the autophagy marker Beclin 1 in fibroblasts obtained from young (*n* = 5) and age-advanced (*n* = 5) study participants. (**A**). TET1 overexpression-related changes in the expression of Beclin 1 and percentage of cells expressing this protein. (**B**). TET2 overexpression-related changes in the expression of Beclin 1 and percentage of cells expressing this protein. Statistical analysis was performed using the Mann-Whitney *U* test (young pSELECT-GFPzeo vs. age-advanced pSELECT-GFPzeo) and Wilcoxon matched-pairs signed rank test (pSELECT-GFPzeo vs. pSELECT-GFPzeo-TET1 or pSELECT-GFPzeo-TET2). MFI—mean fluorescence intensity.

**Figure 4 biomedicines-11-01659-f004:**
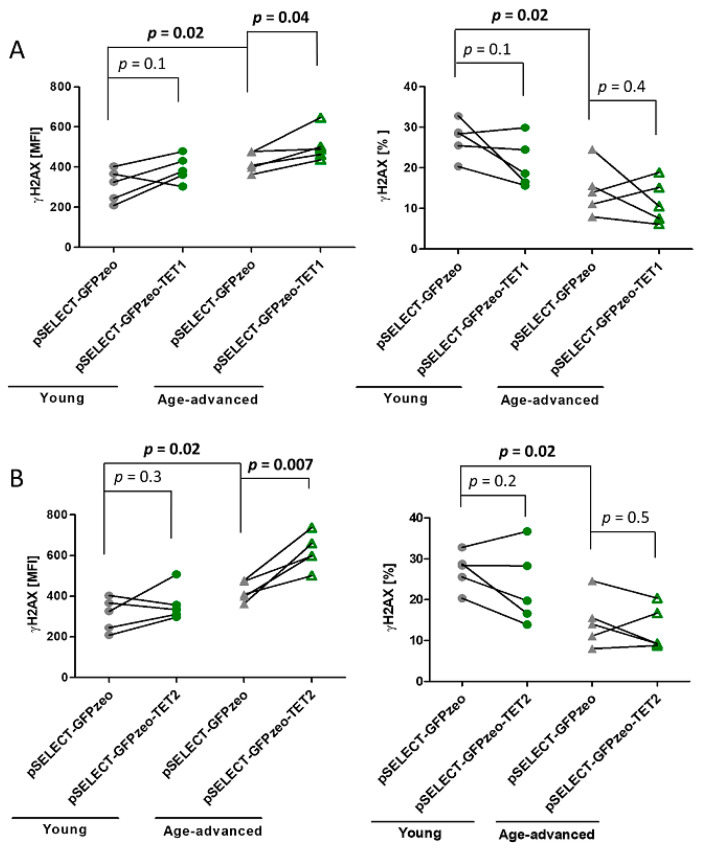
Effects of TET1 and TET2 overexpression on the double-strand break and repair marker γH2AX in fibroblasts obtained from young (*n* = 5) and age-advanced (*n* = 5) study participants. (**A**). TET1 overexpression-related changes in the amount of γH2AX and percentage of cells expressing it. (**B**). TET2 overexpression-related changes in the amount of γH2AX and percentage of cells expressing it. Statistical analysis was performed using the Mann-Whitney *U* test (young pSELECT-GFPzeo vs. age-advanced pSELECT-GFPzeo) and Wilcoxon matched-pairs signed rank test (pSELECT-GFPzeo vs. pSELECT-GFPzeo-TET1 or pSELECT-GFPzeo-TET2). MFI—mean fluorescence intensity.

**Figure 5 biomedicines-11-01659-f005:**
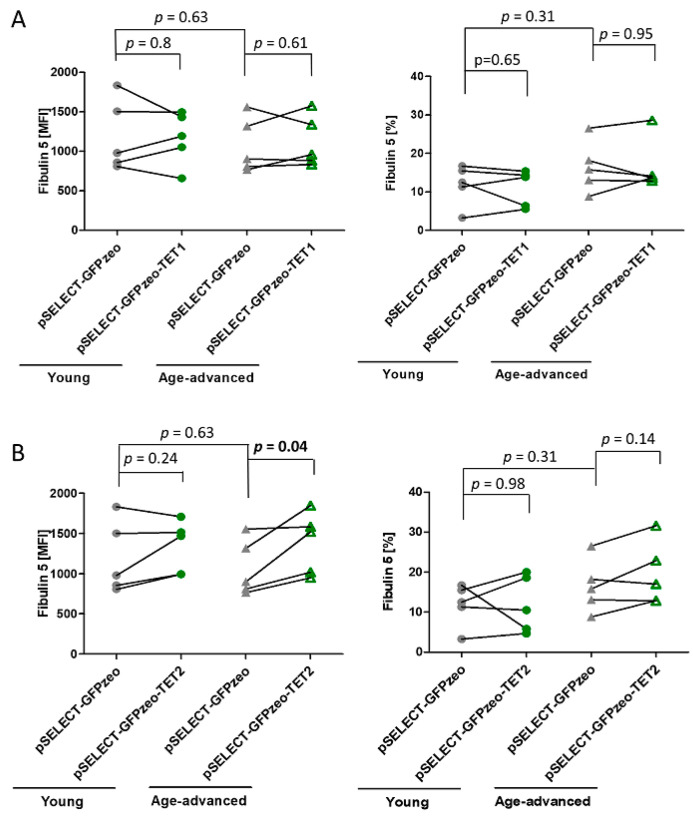
Effects of TET1 and TET2 on Fibulin 5 in fibroblasts obtained from young (*n* = 5) and age-advanced (*n* = 5) study participants. (**A**) TET1 overexpression-related changes in the expression of Fibulin 5 and percentage of cells expressing it. (**B**) TET2 overexpression-related changes in the expression of Fibulin 5 and percentage of cells expressing it. Statistical analysis was performed using the Mann-Whitney *U* test (young pSELECT-GFPzeo vs. age-advanced pSELECT-GFPzeo) and Wilcoxon matched-pairs signed rank test (pSELECT-GFPzeo vs. pSELECT-GFPzeo-TET1 or pSELECT-GFPzeo-TET2). MFI—mean fluorescence intensity.

**Table 1 biomedicines-11-01659-t001:** Summary of the effects of TET1 and TET2 on fibroblasts isolated from the skin of young and age-advanced individuals.

	TET1				TET2			
	Young		Age-Advanced		Young		Age-Advanced	
	Level	%	Level	%	Level	%	Level	%
Proliferation (BrdU incorporation)	-	↓	-	↓	-	↓	-	↓
Apoptosis (Annexin V)	-	↓	-	na	-	na	-	na
Autophagy (Beclin 1)	na	na	↓	na	na	na	na	na
DNA damage and repair (γH2AX)	na	na	↑	na	na	na	↑	na
Fibroblast activity (Fibulin 5)	na	na	na	na	na	na	↑	na
Senescence (p16^INK4A^)	na	na	na	↓	↑	↓	na	↓

↑: increase. ↓: decrease. na: not affected.

## Data Availability

On reasonable request.

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
