# Peer review of "Ten-Eleven Translocation 1 and 2 Enzymes Affect Human Skin Fibroblasts in an Age-Related Manner"

_biomedicines, 2023, doi:10.3390/biomedicines11061659_

Round 1

Reviewer 1 Report

The manuscript entitled “Ten-eleven translocation 1 and 2 enzymes affect human skin fibroblasts in an age-related manner” by Kolodziej-Wojnar et al., studied the role of ten-eleven translocation (TET) enzymes namely TET1 and TET2 in ageing related alterations of human fibroblasts. The effects of TET on proliferation, apoptosis, autophagy, DNA double-strand break (DSB) repair and skin elastic fiber markers were studied to draw the conclusions.

Specific Concerns: 

1.    How TET enzymes impact on the ageing of different organs? Which organs are the most and least effected by TET enzymes along with ageing?

2.     As the efficiency of stable transfection of the fibroblasts are very low with TET over-expression vectors, does the authors consider transient transfection as an alternative?

3.   Figure 1: Does 5hmC co-localize with TET? Scale bar for the microscopic images must be included.

4.   Do the authors observe any change in the DSB repair proteins such as RAD51 foci formation?

5.   A table summarizing the effects of TET1 and TET2 on the age-related changes of fibroblasts might be helpful.

6.     Is there any change in the morphology of the cells with ageing?

7.     Does TET enzymes effect the skin fibroblast composition such as proteoglycans, glycosaminoglycans, collagen etc. with ageing?

8.     Is it possible that the observed effects of TET enzymes on ageing is related to gender bias?

9.  The effects of TET enzymes on ageing highly depend on the developmental stage, tissue type and pathophysiological factors, a tabular representation should be easy to interpret.

10.Reference required for the following statement:

Based on the avaliable data by other authors, we hypothesize that age-related changes in the biological function of TETs are due to changes in the cell’s redox state, availability of cofactors, or a shift in the expression or activities of TET interacting proteins.

11.Reference style must be corrected (reference # 3 needs attention).  

Author Response

REVIEWER 1

We would like to thank the Reviewer for a valuable review. Below are our answers to questions and comments.

  1. How TET enzymes impact on the ageing of different organs? Which organs are the most and least effected by TET enzymes along with ageing?

The following has been added to the discussion section, page 11, lines 304-314: “Notably, epigenetic drift involving aging-associated changes in 5-hydroxymethylation is an inseparable element of growing old. However, even though there are data regarding the aging-related changes in 5hmC (21,32], there is much fewer data regarding the role of TET enzymes in human aging. For example, the expression of TETs in postmortem brains of Alzheimer’s disease sufferers was significantly decreased compared to age-matched, disease-free control brains [33]. TET3 but not TET1 or TET2 protein expression increased with age in human subcutaneous adipose stem cells [21]. A comprehensive study in short-lived killifish Nothobranchius furzeri aging model revealed a substantial reduction in tet3 transcriptional activity in the brain and liver, coexisting with a significant age-associated decrease in 5hmC, while the expression of TET1 and TET2 remained unchanged [34].”

Additional references:

  1. López, V.; Fernández, A.F.; Fraga, M.F. The role of 5-hydroxymethylcytosine in development, aging and age-related diseases. Ageing Res. Rev. 2017, 37, 28–38. doi: 10.1016/j.arr.2017.05.002.
  2. Zhang, Y.; Zhang, Z.; Li, L.; Xu, K.; Ma, Z.; Chow, H.M.; Herrup, K.; Li, J. Selective loss of 5hmC promotes neurodegeneration in the mouse model of Alzheimer's disease. FASEB J. 2020, 34, 16364–16382. doi: 10.1096/fj.202001271R.
  3. Zupkovitz, G.; Kabiljo, J.; Kothmayer, M.; Schlick, K.; Schöfer, C.; Lagger, S.; Pusch, O. Analysis of methylation dynamics reveals a tissue-specific, age-dependent decline in 5-methylcytosine within the genome of the vertebrate aging model Nothobranchius furzeri. Front. Mol. Biosci. 2021, 8, 627143. doi: 10.3389/fmolb.2021.627143.

  1. As the efficiency of stable transfection of the fibroblasts are very low with TET over-expression vectors, does the authors consider transient transfection as an alternative?

Indeed, producing stable TET transfectants is difficult and time-consuming, as it requires many population doublings to expand the stable line. Therefore, such a procedure could result in the disappearance of differences related to the age of the cell donor. In our experiments, we used transiently transfected cells. A low transfection efficiency (5-15%) was, most possibly, due to a very large vector size and the fact that we transfected primary cells.

  1. Figure 1: Does 5hmC co-localize with TET? Scale bar for the microscopic images must be included.

We could not analyze the co-localization of TETs and 5hmC, because 5hmC staining required the use of 4N HCl for antigen retrieval, and such concentration destroys proteins. Scale bars have been added.

  1. Do the authors observe any change in the DSB repair proteins such as RAD51 foci formation?

Unfortunately, we did not perform an analysis examining the change in DSB repair proteins such as RAD51 foci formation. We agree that changes in RAD51 foci formation upon TETs overexpression would provide valuable insights into the DNA hydroxymethylation-related repair mechanisms and their association with aging. While our study was not designed to analyze TET-dependent changes in DNA repair mechanisms in relation to aging, we acknowledge the need for future investigations to explore this aspect further, as DNA repair seems to be altered by the aging process.

  1. A table summarizing the effects of TET1 and TET2 on the age-related changes of fibroblasts might be helpful.

Table 1 has been added, page 11.

  1. Is there any change in the morphology of the cells with ageing?

We did not see changes in the morphology of primary fibroblasts isolated from young and age-advanced individuals. However, cells from younger donors proliferated faster.

  1. Does TET enzymes effect the skin fibroblast composition such as proteoglycans, glycosaminoglycans, collagen etc. with ageing?

We considered performing such experiments but due to a low transfection efficiency, we could not analyze the synthesis and secretion of extracellular matrix components.

  1. Is it possible that the observed effects of TET enzymes on ageing is related to gender bias?

This problem is mentioned as one of the limitations of the study, as we analyzed only female cells, Discussion, page 13, lines 403-407: “Third, we studied only cells originating from females. Nonetheless, we speculate that the effects of TET1 and TET2 in male cells could be similar, as we found no age-related differences in their mRNA and protein expression levels in fibroblasts from males and females [22]. Nevertheless, the potential gender-specific differences in TETs functions in aging need further investigation.”

  1. The effects of TET enzymes on ageing highly depend on the developmental stage, tissue type and pathophysiological factors, a tabular representation should be easy to interpret.

We agree with this statement. However, there are a lot of fragmentary and commonly inconsistent data on this subject obtained in various organisms and tissues. Therefore, in response to this comment, instead of making a table reviewing findings by other authors, we would like to propose the addition of a general statement to the discussion section, page 11, lines 301-304: “Data indicate that TET enzymes regulate the function of embryonic stem cells by maintaining pluripotency and cell fate during development by regulating differentiation [29]. Under physiological conditions, deregulation of the TET enzymes mostly affects stem cells, blood cells, and cells associated with reproduction [30,31].”

Additional references:

  1. Yang, J.; Bashkenova, N.; Zang, R.; Huang, X.; Wang, J. The roles of TET family proteins in development and stem cells. Development 2020, 147, dev183129. doi: 10.1242/dev.183129. 
  2. Huang, G.; Liu, L.; Wang, H.; Gou, M.; Gong, P.; Tian, C.; Deng, W.; Yang, J.; Zhou, T.T.; Xu, G.L.; Liu, L. Tet1 deficiency leads to premature reproductive aging by reducing spermatogonia stem cells and germ cell differentiation. iScience 2020, 23, 100908. doi: 10.1016/j.isci.2020.100908. 
  3. Joshi, K.; Zhang, L.; Breslin, S.J.P.; Kini, A.R.; Zhang, J. Role of TET dioxygenases in the regulation of both normal and pathological hematopoiesis. J. Exp. Clin. Cancer Res. 2022, 41, 294. doi: 10.1186/s13046-022-02496-x. 

  1. Reference required for the following statement: ‘Based on the available data by other authors, we hypothesize that age-related changes in the biological function of TETs are due to changes in the cell’s redox state, availability of cofactors, or a shift in the expression or activities of TET interacting proteins.’

We added:

  1. Salminen, A.; Kauppinen, A.; Kaarniranta, K. 2-Oxoglutarate-dependent dioxygenases are sensors of energy metabolism, oxygen availability, and iron homeostasis: potential role in the regulation of aging process. Cell. Mol. Life Sci. 2015, 72, 3897–914. doi: 10.1007/s00018-015-1978-z. 
  2. Koivunen, P.; Laukka, T. The TET enzymes. Cell. Mol. Life Sci. 2018, 75, 1339–1348. doi: 10.1007/s00018-017-2721-8. 

  1. Reference style must be corrected (reference # 3 needs attention).  

Reference 3 has been corrected. It is a book chapter and its style is in accordance with Biomedicines requirements. We also checked other references.

Reviewer 2 Report

The authors evaluated the effect of TET1 and TET2 overexpression on proliferation and protein expression in human fibroblasts isolated from young and old individuals. The manuscript is well written. However, the authors must provide new information and clarify the following points:

1. The authors must provide an Ethics.

2. The demographic data is the individuals is missing.

3. For future research and reproducibility, the authors must add more precise information from where the skin fibroblasts were obtained.

4. Fig. 1 shows a clear over expression of TET1. Are young or old fibroblasts? They must show data for TET2 as well.  They must show immunofluorescence data for young and old fibroblasts. It is Ok that they show supplementary data with HEK293 as a control. They must add more details to the figure legend. 

On the graphs of Fig. 1: What does MFI stand for?

5. On Fig. 2. It is not clear why the authors paired the data. This comment is applicable to Figs. 3, 4 and 5.

6. On material and methods, 2.6. Indicate the cells used in this protocol.

7. On material & methods: Split the methods described in 2.7. This section is confusing and it is not clear which protocol used fixed or unfixed cells.

8. On material & methods 2.8. They fixed the cells and then used a FACs machine. Please explain.

9. Was the data corrected by transfection efficiency? It could explain the dispersion of the data.

10. On the same vein. Was the dispersion of the data associated with the age of the individuals? Indicate the age of each individual in the graph.

Author Response

REVIEWER 2

We would like to thank the Reviewer for a valuable review. Below are our answers to questions and comments.

  1. The authors must provide an Ethics.

       The ethics statement can be found under the Institutional Review Board Statement and Informed Consent Statement, page 14: “The study was conducted according to the guidelines of the Declaration of Helsinki, and approved by the Bioethics Committee of the Medical University of Warsaw (KB/283/2013). All participants gave written informed consent to participate in the study.”

  1. The demographic data of the individuals is missing.

and

  1. For future research and reproducibility, the authors must add more precise information from where the skin fibroblasts were obtained.

       For the clarity of information, we combined sub-sections 2.1. and 2.2., currently 2.1. Fibroblasts isolation, page 2, lines 79-82: “Healthy skin samples (2 cm2) from a sun-unexposed area of the suprapubic region were collected during elective surgery performed on 5 young (23–35 years) and 5 age-advanced (75–94 years) women without acute inflammation, free of local and systemic diseases possibly affecting the skin's condition such as diabetes, cancer, or autoimmunity.”

  1. 1 shows a clear overexpression of TET1. Are young or old fibroblasts? They must show data for TET2 as well.  They must show immunofluorescence data for young and old fibroblasts. It is Ok that they show supplementary data with HEK293 as a control. They must add more details to the figure legend. 

The experiment shown in Figure 1 was performed on fibroblasts originating from a young donor solely to prove that TET overexpression with our construct results in the most pronounced molecular effect of the action of TET enzymes, namely an increase in 5-hydroxymethylation. Transient transfection did not significantly affect the 5hmC amount, therefore, we decided to produce stable transfectants. Stable transfection is long-term and requires several population doublings to obtain a sufficient number of cells and leads to an attenuation of donor age-related differences (Boraldi et al., Mech Ageing Dev. 2010;131:625-635). Therefore, we did not aim at quantification of this effect in cells originating from young vs. age-advanced donors and, consequently, did not produce 10 lines (5 from young and 5 from age-advanced donors) overexpressing TET1 and 10 overexpressing TET2. Based on the published data by other authors we assumed that overexpression of TET2 will result in the same effect as TET1 overexpression. We added the following sentence, 2.3. Cell transfection, page 3, lines 124-126: “Stable TET1 transfectant line were obtained after two weeks. No TET2 stable transfectants were produced.”

Figure 1 legend has been modified:

“Increased expression of TET1 and amount of 5hmC in primary fibroblasts stably transfected with the pSELECT-GFPzeo-TET1 vector as compared to primary fibroblasts stably transfected with the pSELECT-GFPzeo control vector. GFP: green fluorescent protein, green. TET1: predominantly nuclear localization of TET1, red. 5hmC: nuclear localization of 5-hydroxymethylcytosine, red. Nuclei stained with Hoechst 33342, blue. MFI: mean fluorescence intensity. Scale bar – 50 µm.”

  1. On the graphs of Fig. 1: What does MFI stand for?

MFI: mean fluorescence intensity (description has been added to Figure 1 legend).

  1. On Fig. 2. It is not clear why the authors paired the data. This comment is applicable to Figs. 3, 4 and 5.

As the transfection procedure itself and transfection with foreign DNA can affect various cell functions, we used cells from the same individual transfected with the pSELECT-GFPzeo “empty” vector as an individual-specific control. This is explained in the first paragraph of the 3.2. Effects of TET1 and TET2 on proliferation.

  1. On material and methods, 2.6. Indicate the cells used in this protocol.

Sub-section 2.6. has been supplemented with information regarding cells used (HEK293).

  1. On material & methods: Split the methods described in 2.7. This section is confusing and it is not clear which protocol used fixed or unfixed cells.

We would like to thank the Reviewer for this comment. The 2.6. Proliferation, DNA damage, and apoptosis assays sub-section has been modified, pages 4, lines 157-163: “Proliferation was assayed in fixed cells using the APC BrdU Flow Kit (BD Biosciences, San Jose, CA, USA) and DNA damage was assessed by measuring γH2AX using the Apoptosis, DNA Damage, and Cell Proliferation Kit (BD Biosciences). Forty-eight hours after transfection, cells were stained according to the manufacturer’s protocols. Apoptosis was evaluated in unfixed cells using the PE Annexin V assay (BD Biosciences). Forty-eight hours after transfection, cells were fixed and stained following the manufacturer’s protocol. Next, ….”

  1. On material & methods 2.8. They fixed the cells and then used a FACs machine. Please explain.

Analyses of a given cell function using the FACs method were performed on all cell lines simultaneously. However, it was not possible to grow these cell lines at the same time in sufficient quantities for all such assays. Therefore, after each cell line reached the appropriate, comparable stage of culture, the cells had to be fixed and then frozen. The only exception was apoptosis testing performed on unfixed cells.

  1. Was the data corrected by transfection efficiency? It could explain the dispersion of the data.

Analyses were performed using flow cytometry and only cells successfully transfected with the a. pSELECT-GFPzeo or b. pSELECT-GFPzeo-TET1 or c. pSELECT-GFPzeo-TET2 and expressing GFP (as well as TET1 or TET2 in case of vector b. or c., respectively) were analyzed. The flow cytometry method was used to avoid the problem of potential unequal transfection efficiency indicated by the Reviewer.

  1. On the same vein. Was the dispersion of the data associated with the age of the individuals? Indicate the age of each individual in the graph.

Within the age groups analyzed here (23-35 and 75-94 years), differences were not significantly dependent on age. We would like to ask the Reviewer for accepting the current form of Figures 2-5, as indicating the age of each individual on the graphs will make these figures illegible.

Round 2

Reviewer 1 Report

The manuscript entitled "Ten-eleven translocation 1 and 2 enzymes affect human skin fibroblasts in an age-related manner" by Kolodziej-Wojnar et al., had been revised thoroughly. The queries were addressed in the new version which provides relevant information to the readers. 

Author Response

We would like to thank the Reviewer for her/his work and valuable input into improving our work.

Reviewer 2 Report

The authors responded my queries. They must indicate what type of fibroblasts were used in Suppl. Fig 2; young or old? Ideally, they should show data for young and old fibroblast. If they have the data, they should include this in Suppl. Fig. 2. In any case, this figure shows a difference of ~22% between the transfection of TET1 and TET2. This difference in transfection may contribute  to the differential effect of TET1 and TET2 on aged vs young fibroblast. The authors must discuss this as a limitation of the study.

Author Response

The authors responded my queries. They must indicate what type of fibroblasts were used in Suppl. Fig 2; young or old? Ideally, they should show data for young and old fibroblast. If they have the data, they should include this in Suppl. Fig. 2. In any case, this figure shows a difference of ~22% between the transfection of TET1 and TET2. This difference in transfection may contribute  to the differential effect of TET1 and TET2 on aged vs young fibroblast. The authors must discuss this as a limitation of the study.

We would like to thank the Reviewer for her/his work and valuable input into improving our work. In response to the above comment, we would like to stress that the ~22% difference in transfection efficiency seen on the original supplementary Figure S2 (currently second row on the corrected figure) regarded differences between the pSELECT-GFPzeo “empty” vector and vectors bearing TET1 or TET2 coding sequences. Most possibly, this difference was due to the differences in vector sizes, as the pSELECT-GFPzeo vector is (~4.2 kb, while those encoding the TET proteins are ~10.6 kb (TET1) and ~10.2 kb (TET2). Differences in the transfection efficiency between pSELECT-GFPzeo-TET1 and pSELECT-GFPzet-TET2 were usually only a few percent. We found no consistent differences between transfection efficiency with TET1- and TET2-encoding vectors or age-related differences; the differences were random and varied experiment-to-experiment. Such explanation has been added to the limitations of the study, page 13, lines 403-406: “Third, transfection efficiency was relatively low. However, we found no consistent differences between transfection efficiency with TET1- and TET2-encoding vectors or age-related differences; the differences were random and varied experiment-to-experiment (Supplementary Figure S2).”

Supplementary Figure 2 has been modified following the request. We show plots for young and age-advanced individuals.